# The landscape of metabolic pathway dependencies in cancer cell lines

**James H. Joly** [1¤a], **Brandon T. L. Chew** [1¤b], **Nicholas A. Graham** [1,2,3]*

**1** Mork Family Department of Chemical Engineering and Materials Science, University of Southern California, Los Angeles, California, United States of America, **2** Norris Comprehensive Cancer Center, University of Southern California, University of Southern California, Los Angeles, California, United States of America, **3** Leonard Davis School of Gerontology, University of Southern California, Los Angeles, California, United States of America

¤a Current address: Nautilus Biotechnology, San Carlos, California, United States of America
¤b Current address: Department of Pharmaceutical Sciences and Pharmacogenomics, University of California, San Francisco, California, United States of America
* nagraham@usc.edu

## Abstract

The metabolic reprogramming of cancer cells creates metabolic vulnerabilities that can be therapeutically targeted. However, our understanding of metabolic dependencies and the pathway crosstalk that creates these vulnerabilities in cancer cells remains incomplete. Here, by integrating gene expression data with genetic loss-of-function and pharmacological screening data from hundreds of cancer cell lines, we identified metabolic vulnerabilities at the level of pathways rather than individual genes. This approach revealed that metabolic pathway dependencies are highly context-specific such that cancer cells are vulnerable to inhibition of one metabolic pathway only when activity of another metabolic pathway is altered. Notably, we also found that the no single metabolic pathway was universally essential, suggesting that cancer cells are not invariably dependent on any metabolic pathway. In addition, we confirmed that cell culture medium is a major confounding factor for the analysis of metabolic pathway vulnerabilities. Nevertheless, we found robust associations between metabolic pathway activity and sensitivity to clinically approved drugs that were independent of cell culture medium. Lastly, we used parallel integration of pharmacological and genetic dependency data to confidently identify metabolic pathway vulnerabilities. Taken together, this study serves as a comprehensive characterization of the landscape of metabolic pathway vulnerabilities in cancer cell lines.

## Author summary

Cancer cells rewire their metabolism, which creates targetable metabolic vulnerabilities. Previous analyses of metabolic vulnerabilities in cancer cells have been limited to the analysis of *individual* genes or metabolites. However, metabolic *pathways* exhibit significant cross talk and compensation for one another. We developed a computational method to answer the question: when a metabolic pathway's activity is high, which other metabolic pathways become more essential or less essential? By integrating genetic screen data with

**Data Availability Statement:** Our code is freely available for use at: https://github.com/JamesJoly/MetabolicDependencies.

**Funding:** This work was supported by: 1) The American Cancer Society Grant IRG-16-181-57 (N.

A.G.) https://www.cancer.org/ 2) The 2020 AACR-Bayer Innovation and Discovery Grant, Grant Number 20-80-44-GRAH (N.A.G.). https://www.aacr.org/ 3) The Viterbi School of Engineering at the University of Southern California (N.A.G.). https://viterbischool.usc.edu/ The funders had no role in study design, data collection and analysis, decision to publish, or preparation of the manuscript.

**Competing interests:** The authors have declared that no competing interests exist.

drug response data from FDA approved drugs, we identified cancer cell line dependence on metabolic *pathways* as opposed to individual *genes*. For example, we found that identifying key regulators of metabolic pathways, such as the Pentose Phosphate Pathway, may serve as a biomarker to identify which patients may benefit from antifolate chemotherapies (e.g. methotrexate, 5-fluorouracil). The efforts outlined here serve as the first characterization of the landscape of metabolic pathway vulnerabilities in cancer cell lines. Our results demonstrate the benefit of analyzing dependencies on metabolic pathways as opposed to metabolic genes.

## Introduction

The reprogramming of cellular metabolism was one of the earliest discovered hallmarks of cancer [1]. Cancer cells rewire their metabolism to satisfy the bioenergetic, biosynthetic, and redox demands of tumors. In turn, these metabolic adaptations create cancer-specific vulnerabilities that can be therapeutically targeted [2]. Much research has focused on how individual mutations or DNA copy number alterations reprogram tumor metabolism and create therapeutic opportunities [3–7]. For example, leukemias and gliomas with mutations in isocitrate dehydrogenase (IDH) are sensitive to inhibitors specific to mutant IDH [8,9]. In addition, some tumors are sensitive to depletion or restriction of amino acids, including cysteine in tumors with deleted methylthioadenosine phosphorylase (*MTAP*) [10], serine in tumors with deleted p53 (*TP53*) [11], asparagine in leukemias with low expression of asparagine synthetase (*ASNS*) [12], asparagine in tumors with inhibition of the electron transport chain [13], and methionine in RAS-driven tumors [14]. Other efforts have identified metabolic vulnerabilities related to redox balance including in cancer cells with dysregulated PI3K/AKT signaling [15] or overexpression of the L-glutamate/L-cystine antiporter SLC7A11 [16,17].

While these studies have proved fruitful for advancing the therapeutic targeting of cancer metabolism [2], they have traditionally been limited to the study of individual genes. However, metabolic pathways consist of multiple enzymes which collectively regulate metabolic flux. Furthermore, metabolic pathways often have similar byproducts (e.g. ATP, NAD(P)H), suggesting that pathways may compensate for one another under different contexts. For example, NADPH can be produced by three different pathways: the oxidative pentose phosphate pathway, serine-driven one-carbon metabolism, and malic enzyme [18]. These three pathways can presumably be up- and down- regulated to respond to changes in activity of the other pathways. Since these compensatory mechanisms exist across human metabolism, studying the effects of individual genes may not reflect cancer cell metabolic vulnerabilities at the pathway level. As a result, our understanding of cancer cell dependency on metabolic pathways remains incomplete.

Recent developments in large scale CRISPR-based genetic [19,20] and pharmacologic screening [21] along with large panels of comprehensively characterized cancer cell lines [22] have proved powerful tools for identification of genes essential for cancer cell survival [23], elucidation of drug mechanism-of-action [19,24,25], and discovery of novel candidate drug targets [26,27]. Furthermore, parallel integration of both pharmacologic and gene loss-of-function data has been used to identify drug mechanism(s) of action [25,28–30]. While these databases have served as a rich resource to explore individual gene vulnerabilities and drug sensitivities, there exists a need to probe these datasets on the pathway level.

Here, we aimed to identify cancer cell dependencies on metabolic pathways rather than individual metabolic genes. To do so, we used gene expression data from the Cancer Cell Line

Encyclopedia (CCLE) to infer metabolic pathway activity and then integrated these pathway activities with data from genetic and pharmacologic screens across hundreds of cell lines. We show that this approach provides a comprehensive characterization of the landscape of metabolic pathway dependencies in cancer cell lines. In addition to demonstrating the context-specific nature of metabolic pathway dependence, we identified both known and novel metabolic vulnerabilities, discovered robust associations between drug response and metabolic pathway activity, and independently found metabolic pathway essentialities in both genetic and pharmacological screens. Collectively, we present an approach to integrate gene expression, gene dependency, and drug response data to identify cancer cell dependencies on metabolic pathways.

## Results

### Genetic pathway dependency enrichment analysis identifies metabolic pathway dependencies in genetic screens

To identify metabolic pathway dependencies, we analyzed gene expression data and CRISPR-Cas9 loss-of-function screens from 689 cancer cell lines overlapping between the Cancer Cell Line Encyclopedia (CCLE) [22,26] and the Cancer Dependency Map [20]. First, we inferred metabolic pathway activity for each cell line using single-sample gene set enrichment analysis (ssGSEA) of the RNAseq data from each cell line [31]. To focus on metabolism, we queried 69 metabolic pathways from the Kyoto Encyclopedia of Genes and Genomes (KEGG) [32]. We chose to focus on pathways that are expressed in human metabolism and had more than five genes in each set. Because metabolism is influenced by culture type [33] and culture medium [23], we first divided cancer cell lines by culture type (e.g., adherent v. suspension culture) and media (e.g., RPMI v. DMEM) (**Fig 1A**). Cell lines without annotations for either of these features were removed, leaving 300 adherent cell lines cultured in RPMI, 153 adherent cell lines cultured in DMEM, 66 suspension cell lines cultured in RPMI, and 2 suspension cell lines cultured in DMEM. Since the number of suspension cell lines was small, we focused our analysis on adherent cell lines. The resulting ssGSEA normalized enrichment scores (ssNESs) represent the metabolic pathway activity relative to all other cell lines within the respective cell culture medium. Next, we correlated the cell line-specific NESs for each metabolic pathway with cell fitness effects from CRISPR-Cas9 loss-of-function screens (16,643 gene knockouts). Here, each correlation coefficient represents the association between metabolic pathway activity and gene essentiality, with positive values representing increased gene dependency in cell lines with increased metabolic pathway activity. Conversely, a negative correlation indicates increased gene dependency in cell lines with decreased metabolic pathway activity. Finally, to measure the essentiality of the entire metabolic pathway, as opposed to individual genes, we then ranked the resulting 16,643 correlation coefficients and analyzed the rank list using GSEA again querying the KEGG metabolic pathways. Here, positive NES values represent increased pathway dependency upon increased pathway activity, whereas negative NES values represent increased pathway dependency upon decreased pathway activity. Because this approach integrates the essentiality of all genes across a metabolic pathway into a single metric of pathway dependency, we termed this approach genetic pathway dependency enrichment analysis (Genetic PDEA).

To analyze the sensitivity of our Genetic PDEA approach, we analyzed simulated gene expression and gene dependency data using the pipeline outlined in Fig 1A. Gene expression data (16,643 genes) was simulated for 300 cell lines using a normal distribution for each cell line ($\mu = 0$, $\sigma = 0.5$) which reflects the shape of CCLE gene expression data (**S1 Fig**). Then, a synthetic gene set of 25 genes was perturbed using a normal distribution gradient. In cell line

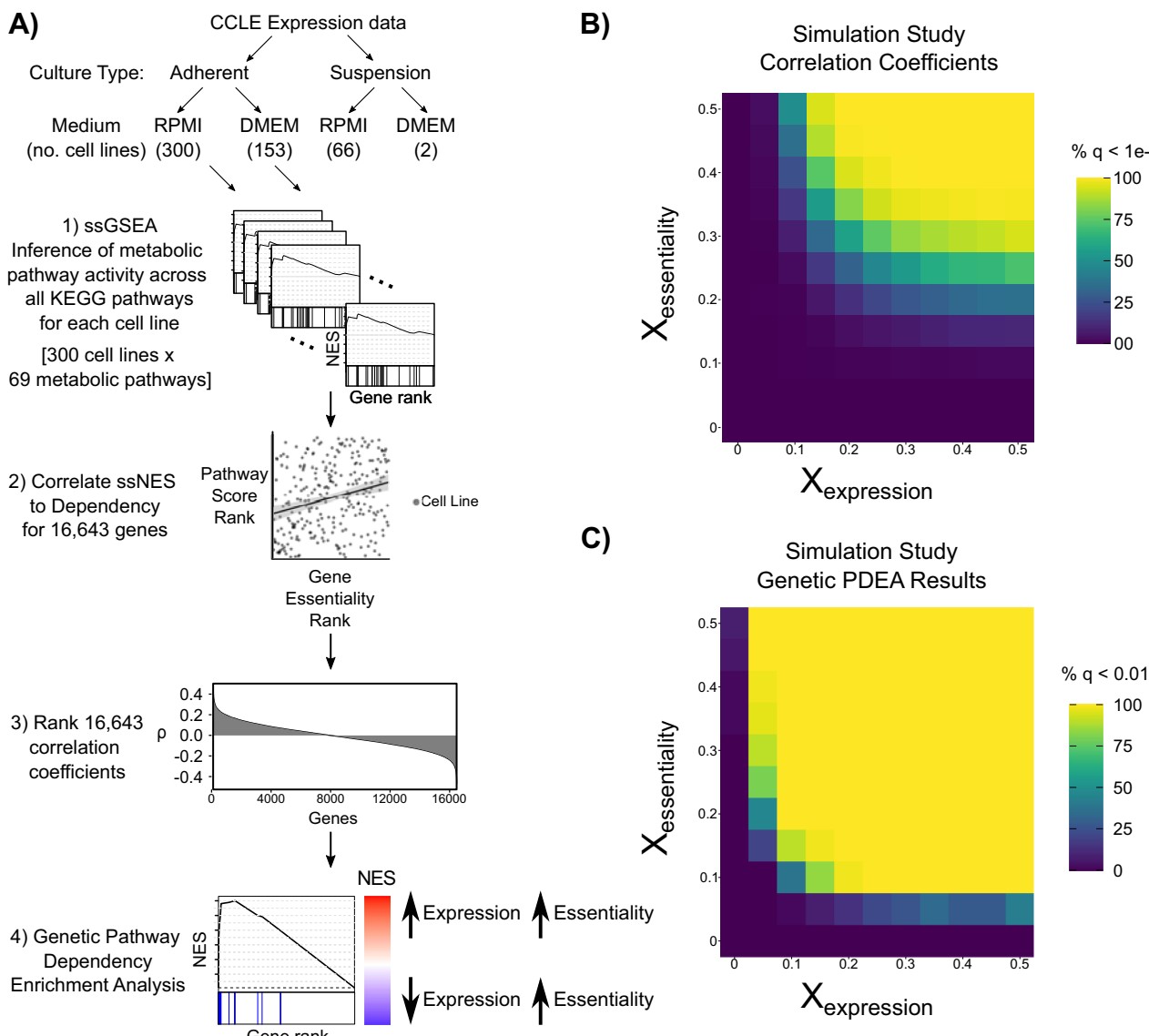

**Fig 1. Integration of gene expression and CRISPR gene dependencies to identify metabolic pathway dependencies. A)** Schematic outlining the approach for Genetic Pathway Dependency Enrichment Analysis (Genetic PDEA). Cancer cell lines from the CCLE were first stratified by culture type (adherent, suspension) and culture medium (RPMI, DMEM), and then their metabolic pathway activity was inferred using single-sample GSEA (ssGSEA). The resulting pathway activities were integrated with gene dependency to assess association with metabolic pathway activity. **B-C)** Simulated data (see Methods) was used to assess the sensitivity of the Genetic PDEA approach. The heatmaps represent the percentage of significant results at each gradient added. Values added to the expression gradient resulted in slightly stronger correlation coefficients and Genetic PDEA results compared to values added to dependency gradient.

1, the 25 genes were replaced with values from a normal distribution with μ = -X, σ = 0.5, and in cell line 300, the 25 genes were replaced with values from a normal distribution with μ = +X, σ = 0.5. For cell lines 2–299, the 25 genes were replaced with values from normal distributions with μ sequentially increasing from -X to X. ssGSEA NESs were then calculated for the synthetic gene set for all 300 cell lines. Next, gene dependency data was simulated for the same 300 cell lines using a similar normal distribution gradient method. For both gene expression and gene dependency data, values for the perturbation X were varied from 0 to 0.5. Then, Spearman correlation coefficients between synthetic gene set activity (ssGSEA NESs) and gene

dependency were calculated for all 16,643 genes. Finally, GSEA was run to calculate the simulated Genetic PDEA values as outlined in Fig 1A. Analyzing 50 replicates of this simulation pipeline, we found that both the correlation coefficients and Genetic PDEA NES results were more strongly influenced by expression gradients added than dependency gradients added (**Fig 1B and 1C**). This suggests that our Genetic PDEA approach to identify metabolic pathway vulnerabilities in cancer cells is more sensitive to changes in metabolic pathway activity than changes in gene dependency. However, when the perturbation X was large for the dependency gradient and small for the expression gradient (or vice versa), significant Genetic PDEA NES values were still obtained.

Having validated the sensitivity of our Genetic PDEA approach, we next investigated cancer cell line dependency across all KEGG metabolic pathways. We tested a total of 4,692 metabolic pathway combinations and identified 190 significant enrichments (FDR < 0.05) between metabolic pathway activity and pathway dependency in the adherent RPMI data set and an additional 190 significant enrichments in the adherent DMEM data set (**S1 Table**). Leading edge genes for each significant pathway dependency are listed in **S1 Table**. We next clustered Genetic PDEA NES values across all pathway activities (columns) and pathway dependencies (rows) (**Figs 2A and S2A**) and found that related pathways clustered together on the x-axis (pathway activity) but not on the y-axis (pathway dependency). This suggests that pathway activity but not pathway dependency is similar within a group of related pathways. For example, glycan biosynthesis pathways exhibit similar correlations with activity of other glycan pathways (columns) but differ in their own dependency (rows). We also found that the no single metabolic pathway had universally positive or negative NES values, suggesting that cancer cells are not universally dependent on any metabolic pathway. Interestingly, the strongest metabolic dependency of adherent RPMI cancer cell lines was Folate Biosynthesis (hsa00790) when One-Carbon Pool by Folate (hsa00670) pathway activity was high (**Fig 2B**). This enrichment was driven by the genes *QDPR*, *ALPI*, *ALPP*, and *GCH1*. Notably, Folate Biosynthesis is directly upstream of One-Carbon Pool by Folate. Furthermore, one of the strongest Genetic PDEA results in both the adherent DMEM and adherent RPMI analyses recapitulated a link between increased dependency on One-Carbon Pool by Folate in cells with increased TCA cycle activity [34] (**Figs 2C and S2B**). Additionally, some metabolic pathways exhibited context-specific dependencies. For example, the dependency on the TCA Cycle (hsa00020) was increased in adherent RPMI cell lines with decreased Glycolysis-Gluconeogenesis (hsa00010) activity, whereas dependency on the TCA cycle was increased in adherent RPMI cell lines with increased Pentose Phosphate Pathway (hsa00030) activity (**S3 Fig**). This new finding suggests that the diversion of glucose from glycolysis to the pentose phosphate pathway may confer increased dependency on the TCA Cycle. We next asked whether there existed a general relationship between a metabolic pathway's activity and its own essentiality. Of the 69 metabolic pathways queried, 36 had a negative NES and 33 had a positive NES for adherent RPMI cell lines (**S4A Fig** and **S2 Table**). A similar distribution was observed in DMEM, although the pathways with positive and negative NES values were not the same as in RPMI (**S4B Fig**). While there was not general agreement between the self-dependencies, one gene set did exhibit significant Genetic PDEA NES for both Adherent RPMI and Adherent DMEM cells. Specifically, Riboflavin Metabolism (hsa00740) exhibited significant negative Genetic PDEA NES, indicating that when Riboflavin Metabolism activity is low, the dependency on Riboflavin Metabolism genes increases. Taken together, these results suggest that there is no generic rule regarding a metabolic pathway's activity and its essentiality. Rather, these results indicate that metabolic pathway dependency is highly context specific such that metabolic pathway activity influences metabolic pathway essentiality.

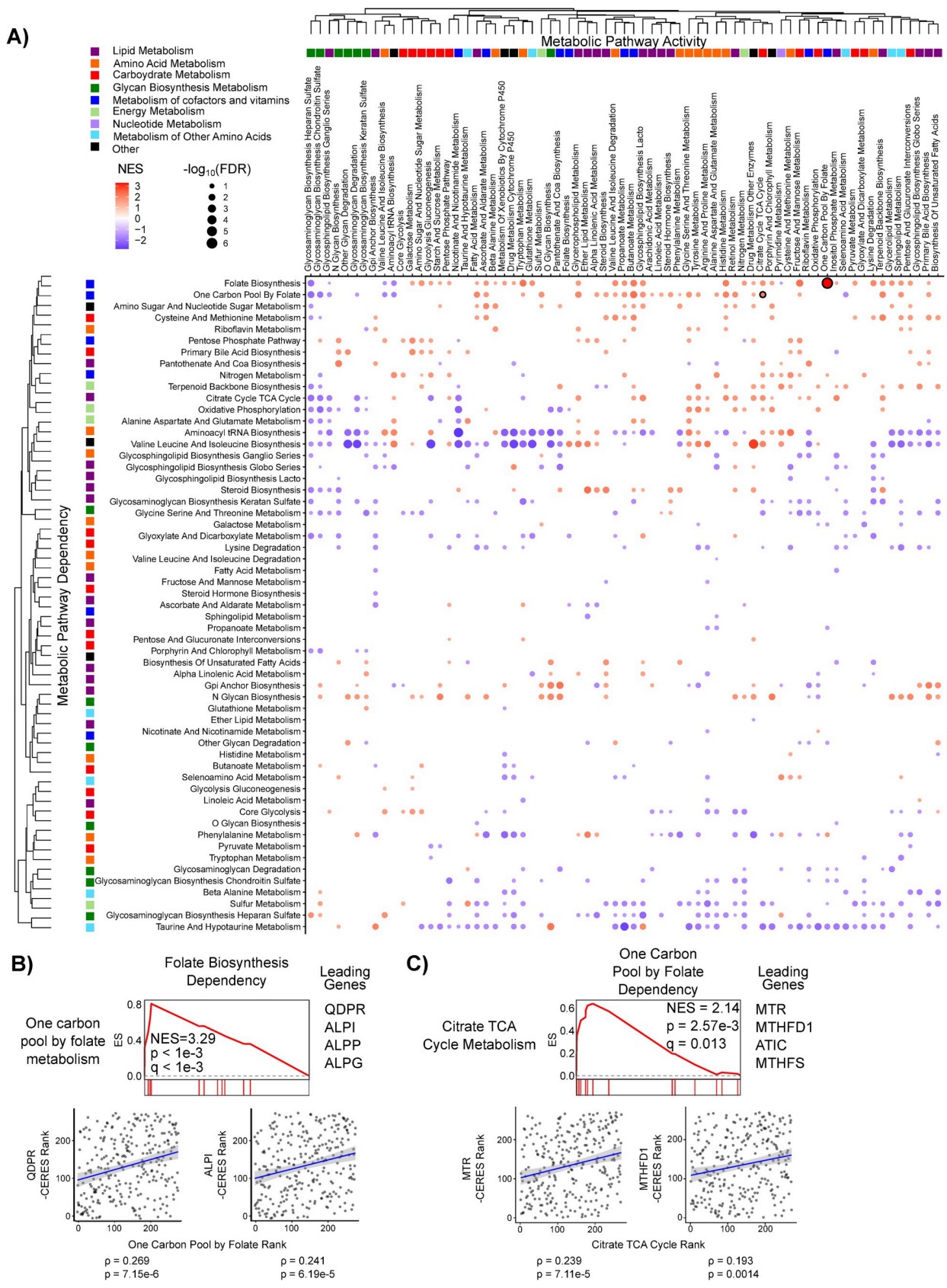

**Fig 2. Global analysis of metabolic dependency data reveals context-specific pathway essentialities. A)** Metabolic pathway activity was inferred using ssGSEA for 300 adherent cell lines cultured in RPMI and correlated to gene dependency data from The Cancer Dependency Map (DepMap). Correlation coefficients were then ranked and Genetic Pathway Dependency Enrichment Analysis (Genetic PDEA) was run using the KEGG metabolic pathways (see Fig 1). Hierarchical clustering was performed on the Genetic PDEA normalized enrichment scores (NES). Results for pathways with FDR < 0.25 are plotted. Dots are colored according to their NES and sized according to the -log$_{10}$ of the false discovery rate (FDR). Numerical values for each pathway can be found in S1 Table. Results shown in B and C are highlighted with a black outline. **B)** Cancer cell dependency on Folate Biosynthesis (hsa00790) was increased when One-Carbon Pool by Folate (hsa00670) pathway activity was high. The scatter plots of pathway activity NES and gene dependency (-CERES) for leading-edge genes *QDPR* and *ALPI* are shown. **C)** Dependency on One-Carbon Pool by Folate metabolism (hsa00670) is increased when TCA cycle (hsa00020) activity is increased. The scatter plots of pathway activity NES and gene dependency (-CERES) for leading-edge genes *MTR* and *MTHFD1* are shown.

## Validation of genetic PDEA results using other CRISPR data sets and pathway activity metrics

We next wanted to examine whether these findings were reproducible across data sets and methods. To examine whether these findings were reproducible, we analyzed data from another large-scale pan-cancer CRISPR-Cas9 gene dependency data set (Sanger Institute) [35] using the Genetic PDEA pipeline. Since the underlying gene expression profiles were derived from the CCLE for both Sanger and DepMap Genetic PDEAs, we combined statistical tests using the harmonic mean p-value (HMP) which combines dependent statistical tests while controlling the family-wise error rate (FWER) [36]. Applying an HMP threshold of 0.05, we found that 96% and 90% of significant results in the DepMap Genetic PDEA were recapitulated in the combined DepMap and Sanger Genetic PDEA results for Adherent RPMI and Adherent DMEM cells, respectively (**S5 Fig** and **S3 Table**).

Next, we tested the effect of metrics other than ssGSEA for inference of metabolic pathway activity. We chose to use normalized weighted average expression (NWAS), which accounts for overlap between gene sets [37]. To directly compare NWAS with ssGSEA, we re-ran our pipeline using NWAS to analyze dependency on the Pentose Phosphate Pathway which contains several enzymes that are shared with other metabolic pathways (e.g., PRKL, PFKM, and PKFP are present in both Glycolysis-Gluconeogenesis and Pentose Phosphate Pathway gene sets). We found broad agreement between the metabolic pathway dependencies when using either NWAS or ssGSEA for both Adherent RPMI and Adherent DMEM cell lines (Spearman r of 0.606 and 0.691, respectively) (**S6 Fig**). Taken together, these findings demonstrate that our Genetic PDEA results for metabolic pathway dependencies are reproducible across different data sets and methods.

## Media composition influences metabolic pathway dependency

We next investigated whether cancer cell line metabolic pathway dependencies were influenced by media composition. To assess the essentiality of a pathway, we weighted each NES from Genetic PDEA by its -log$_{10}$ FDR and then took the mean of all weighted NESs. We found striking differences between DMEM and RPMI metabolic pathway essentialities that can be partly explained by media composition (**Fig 3**), consistent with the finding that the essentiality of individual metabolic genes is influenced by culture medium [23]. For example, cancer cells cultured in RPMI exhibited a strongly positive average weighted NES for Folate Biosynthesis (hsa00790) whereas cancer cells cultured in DMEM did not. Notably, DMEM contains four times the concentration of folate (4 mg/L) compared to RPMI (1 mg/L), suggesting that cancer cells grown in DMEM need to synthesize less folate, thereby reducing their dependency on Folate Biosynthesis. Similarly, cancer cells grown in DMEM were more dependent on Oxidative Phosphorylation (hsa00190) than cancer cells grown in RPMI. One function of oxidative phosphorylation is to enable aspartate synthesis to accept electrons from the electron transport chain [38,39]. Since RPMI and DMEM contain 150 μM and 0 μM aspartate, respectively, the

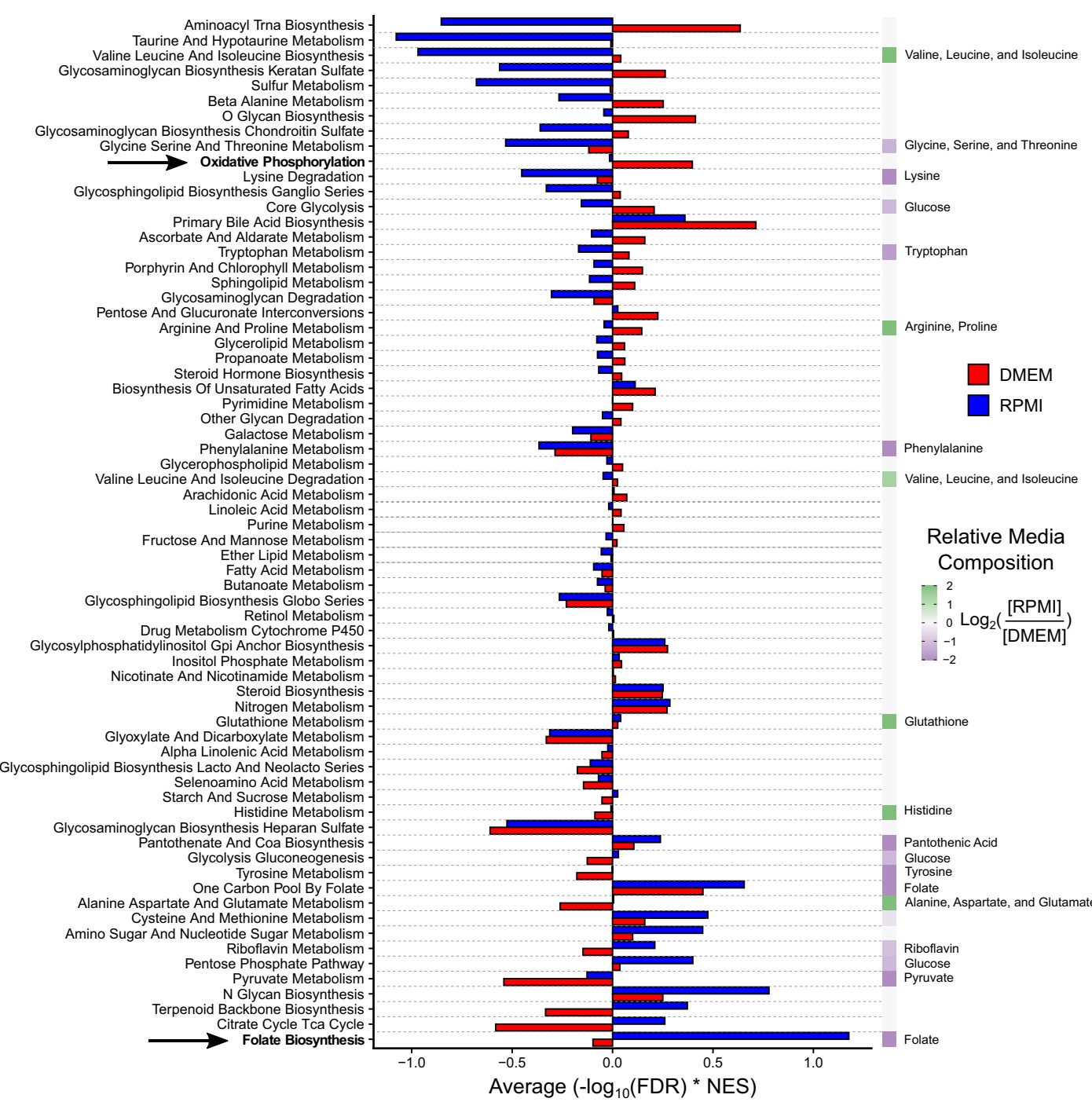

**Fig 3. Media composition influences metabolic pathway dependency.** For adherent cancer cell lines cultured in RPMI (Fig 2) and DMEM (S2 Fig), the metabolic pathway dependency NESs from Genetic PDEA analysis were weighted by -log$_{10}$ FDR. The weighted NESs were then averaged across all 69 KEGG metabolic pathways. Pathways are ranked by the difference between DMEM and RPMI. The relative media composition between RPMI and DMEM are shown on the right on a purple to green heat map with the relevant metabolite(s) indicated. For pathways with multiple metabolites, the average of the metabolites was taken. For example, the concentration of folate in RPMI and DMEM is 1 mg/L and 4 mg/L, respectively. Folate is shown twice because it is both the product of Folate Biosynthesis and the input to One-Carbon Pool by Folate. The dependency on Folate Biosynthesis was much higher in RPMI than in DMEM because these cells must synthesize more folate. Conversely, the dependency on oxidative phosphorylation is much higher in DMEM. This may be due to differences in aspartate levels (RPMI 150 μM, DMEM 0 μM). The indicated pathways are highlighted in bold. Overall, pathways that contained a metabolite which is differentially abundant in cell culture media exhibited a significant difference in pathway essentiality in DMEM and RPMI (p = 3.9x10$^{-4}$ by paired Mann-Whitney U test). In contrast, pathways that do not contain differentially abundant cell culture media metabolites did not exhibit significantly different pathway dependencies (p = 0.545 by paired Mann-Whitney U test).

increased dependency on oxidative phosphorylation in DMEM may reflect an increased need for aspartate synthesis. Interestingly, we did not observe a strong difference in dependency on Aspartate, Alanine, and Glutamate metabolism (hsa00250) between RPMI and DMEM despite the difference in aspartate concentrations. We posit that the inclusion of alanine and glutamate metabolism genes may be a confounding factor for analyzing aspartate dependency.

To test whether media composition globally affected metabolic pathway abundance, we divided metabolic pathways into two groups: 1) pathways that contain metabolites with differential abundance in cell culture media (e.g., Glycolysis-Gluconeogenesis (glucose), Folate biosynthesis (folate)); and 2) pathways that do not contain metabolites with differential abundance in cell culture media (e.g., fatty acid metabolism, fructose and mannose metabolism). We then tested whether the metabolic pathway dependency in RPMI and DMEM differed for both groups of pathways. Notably, we found that pathways that contain differentially abundant metabolites in cell culture media exhibited significantly different pathway essentialities in DMEM and RPMI ($p = 3.9 \times 10^{-4}$). In contrast, pathways that do not contain metabolites that are differentially abundant in cell culture media did not exhibit significantly different pathway dependencies in DMEM and RPMI ($p = 0.545$). Taken together, these results suggest that media composition influences cancer cell line metabolic pathway dependency and that future studies of metabolic vulnerabilities should take media composition into consideration.

## Metabolic pathway activity is correlated with anti-cancer drug sensitivity

We next sought to integrate metabolic pathway activity with large scale pharmacologic screens. We therefore used the PRISM drug repurposing database [21], which contains 1,448 compounds screened against 499 cell lines at 8 different doses. Once again, cell lines were separately processed by culture type and culture medium with a focus on adherent cell lines. Compounds measured in less than 150 cell lines were removed from the analysis, leaving 1,390 compounds. We then correlated drug response area-under-the-curve (AUC) with metabolic pathway activity after multiplying the response of drugs classified as positive regulators (e.g., agonists) by -1 for directional consistency (**Fig 4A**). Here, the AUC represents the dose dependent effect of a drug on cell growth, with a lower AUC representing a stronger response to drug [21]. First, we asked whether there were any drugs with directional agreement across the RPMI and DMEM analyses. From a possible 101,360 possible drug:metabolic pathway combinations, 66 combinations passed FDR-corrected significance thresholds ($q < 0.05$) and were of the same sign in both RPMI and DMEM (**S4 Table**). Notably, zero results that passed FDR correction were of different sign. Many of the common associations in RPMI and DMEM were tyrosine kinase inhibitors (TKIs), which have been extensively linked to metabolism [40]. Interestingly, we found a strong association between decreased Core Glycolysis (hsa_M00001) pathway activity and increased sensitivity to AZD8931, an inhibitor of EGFR and ERBB2 (HER2) (**Fig 4B**). We also found a strong association between increased α-linoleic acid metabolism (hsa00592) and sensitivity to afatinib, another EGFR inhibitor (**Fig 4C**). Of the non-TKI results, we found a link between decreased phenylalanine metabolism (hsa00360) and increased sensitivity to atorvastatin, an HMGR inhibitor (**Fig 4D**). HMGR is the rate limiting enzyme in the cholesterol biosynthetic pathway [41] and multiple reports have suggested that elevated levels of phenylalanine inhibit cholesterol biosynthesis [42–44]. Increased response to atorvastatin when phenylalanine metabolism activity is low suggests that decreased phenylalanine metabolism and HMGR inhibitors may be redundant. Lastly, we found a known link between decreased mucin type O-glycan biosynthesis pathway (hsa00512) activity and increased sensitivity to the HSP90 inhibitor NMS-E973 [45] (**Fig 4E**). Taken together, these results indicate that metabolic pathway activity can be associated with anti-cancer drug sensitivity independent of cell culture medium.

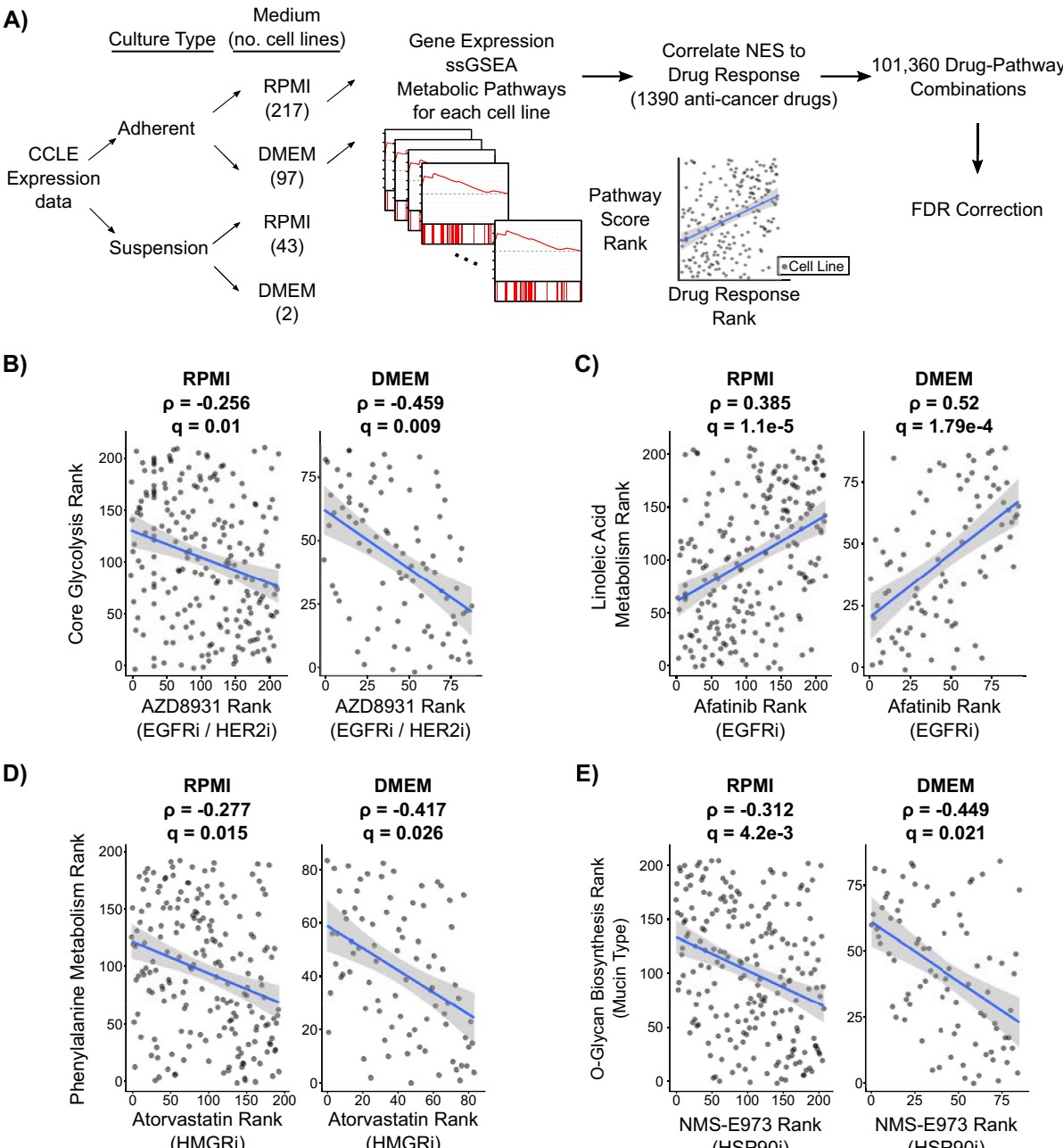

**Fig 4. Metabolic pathway activity identifies anti-cancer drug sensitivity independent of cell culture medium. A)** Schematic representing the strategy used to integrate metabolic pathway activity with drug response screens. Cancer cell lines were separately processed by culture type and culture medium with a focus on adherent cell lines. All correlation p-values were FDR corrected using a Benjamini-Hochberg correction. **B-E)** Scatter plots of significant drug:metabolic pathway combinations (FDR < 0.05) in both DMEM and RPMI mediums. Correlation coefficients and FDR corrected p-values are shown for each correlation. The annotated gene target of each drug is listed below the drug name. The remaining significant associations are listed in S2 Table.

## Pharmacological pathway dependency enrichment analysis reveals common metabolic pathway vulnerabilities

Having identified strong associations between metabolic pathway activity and individual drugs, we next asked whether there were commonalities in the response of cancer cell lines to

families of metabolic pathway inhibitors. Drugs were mapped to metabolic pathways using their annotated target(s) and grouped according to the KEGG metabolic pathways database. Drugs were also classified as positive or negative regulators based on their annotated mechanism of action. For example, drugs labeled as "agonists" or "activators" were classified as positive regulators whereas "blockers" and "antagonists" were classified as negative regulators. To enable a consistent pathway analysis, correlation coefficients for positive regulators were multiplied by -1. Pathways with less than 4 drugs were omitted from the analysis, leaving 46 sets of drugs targeting metabolic pathways. We then analyzed the enrichment of these metabolic pathway inhibitors in the rank list of drug sensitivity-metabolic pathway activity correlation coefficients (n = 1,390). We termed this approach Pharmacological Pathway Dependency Enrichment Analysis (Pharmacological PDEA) (**S7A Fig**). To test the sensitivity of this approach, we again performed a simulation study with simulated gene expression data and simulated drug sensitivity (1,390 drugs). This approach was analogous to the simulation study of Fig 1B with 1,390 drug sensitivities replacing 16,643 gene dependencies. Like Genetic PDEA, we found that expression gradients resulted in stronger results than dependency gradients for both individual drug correlation coefficients and Pharmacological PDEA (**S7B and S7C Fig**).

We then clustered the pharmacological PDEA NESs and found that pathways with similar function clustered together based on pathway activity (columns) more than with their dependency profile across pathways (rows) (**Figs 5A and S8A and S5 Table**), similar to our results with Genetic PDEA (**Fig 2**). For example, the activities of Core Glycolysis (hsa_M00001), Fructose and Mannose Metabolism (hsa00051), Starch and Sucrose Metabolism (hsa00500), and Pentose Phosphate Pathway (hsa00030) were clustered together in the adherent RPMI cell lines. Among the strongest Pharmacological PDEA results, we found that sensitivity to inhibitors of Terpenoid Backbone Biosynthesis (hsa00900) was increased in adherent RPMI cancer cells with high Alanine, Aspartate, and Glutamate Metabolism (hsa00250) (**Fig 5B**). We also found an interesting link between decreased pentose phosphate pathway (PPP) pathway activity and increased sensitivity to folate biosynthesis inhibitors (**Fig 5C**). Because folate biosynthesis inhibitors prevent the generation of NADPH via one-carbon metabolism, these inhibitors may be more damaging to cellular redox balance when PPP expression is low. In fact, most strong results for inhibitors of folate biosynthesis occur when overall metabolic pathway activity is low (**Figs 5D and S8B**). Conversely, inhibitors of Ascorbate and Aldarate metabolism (hsa00053) are more effective when overall metabolic pathway activity is high (**Fig 5E**). This may be because ascorbate (also known as Vitamin C) is an effective antioxidant used to detoxify reactive oxygen species (ROS). ROS are a byproduct of many metabolic reactions such as oxidative phosphorylation and methionine metabolism, which indirectly produces ROS by supporting polyamine synthesis [10]. Indeed, the sensitivity to inhibitors of Ascorbate and Aldarate metabolism is stronger when expression of these ROS producing pathways is high (**Fig 5F**) suggesting that ascorbate's role as an antioxidant is crucial in this context. Taken together, these results reveal contexts in which pharmacological inhibition of metabolic pathways results in decreased cell survival.

## Integration of pharmacologic and genetic screens reveals consistent metabolic vulnerabilities

Next, we sought to integrate results from genetic and pharmacological screen data to identify consistent metabolic pathway dependencies found independently in both analyses. First, we integrated individual gene dependency correlations with their corresponding drug sensitivity correlations by first annotating each drug with its gene target(s). We then summed the gene

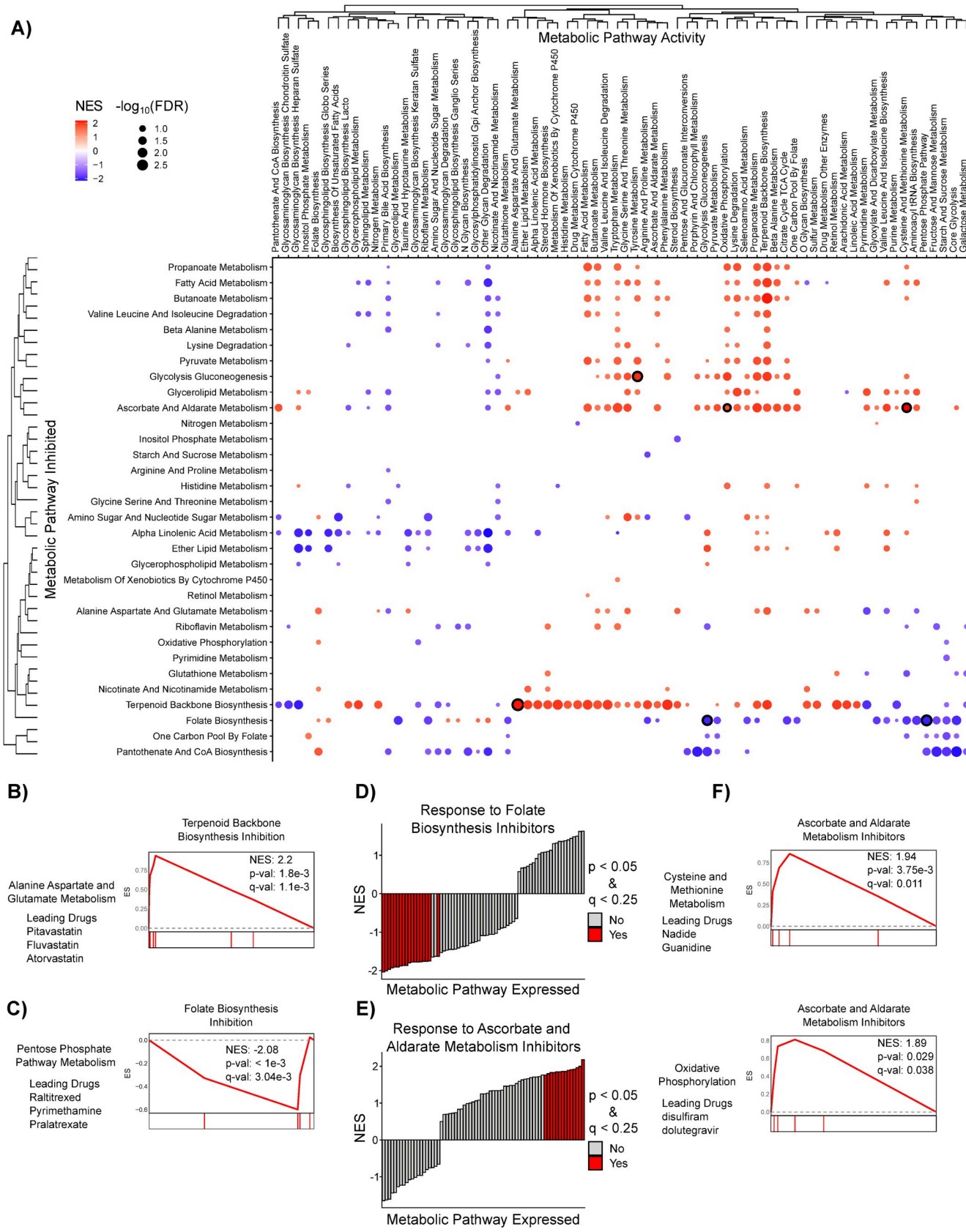

**Fig 5. Pharmacological PDEA reveals consistent metabolic pathway vulnerabilities in Adherent RPMI cell lines. A)** Pharmacological PDEA (S4 Fig) was performed on 1,390 anti-cancer drugs from the PRISM database. Drugs were mapped to metabolic pathways by their annotated target(s) and then the enrichment of these metabolic pathway inhibitors was analyzed in the rank list of drug sensitivity-metabolic pathway activity correlation coefficients. Hierarchical clustering was performed on NES values, and results with FDR < 0.25 are plotted. Dots are colored according to the NES and sized according to the -$\log_{10}$ FDR. Dots with black outline correspond to results shown in panels B-C and F. **B-C)** Increased Alanine, Aspartate, and Glutamate metabolism (hsa00250) correlates with increased response to inhibitors of terpenoid backbone biosynthesis. In contrast, decreased Pentose Phosphate Pathway metabolism correlates with increased response to inhibitors of Folate Biosynthesis (hsa00790). **D-E)** Inhibitors of Folate Biosynthesis (hsa00790) are more effective when overall metabolic pathway expression is low, whereas inhibitors of Ascorbate and Aldarate Metabolism (hsa00053) are more effective when overall metabolic pathway expression is high. **F)** Representative mountain plots and the drug(s) driving enrichment of metabolic pathway activities that strongly correlate with response to inhibitors of Ascorbate and Aldarate metabolism are shown.

dependency correlation coefficient and the drug sensitivity correlation coefficient for each drug target and assessed significance by a permutation test with FDR correction (**Fig 6A**). Out of 187,818 gene+drug:metabolic pathway combinations, we found 176 results that passed an FDR-corrected significance threshold of 0.01 (**S6 and S7 Tables**). Interestingly, all significant results targeted known cancer driver genes such as *EGFR*, *HER2*, *PIK3CA*, and *BRAF* (**Fig 6B–6E**). These results included a known interaction between HER2 inhibitors and retinol metabolism, whereby increased retinol metabolism enhances sensitivity to HER2 inhibition [46] (**Fig 6C**). Additionally, some results identified well known molecular interactions, such as *BRAF* and *PIK3CA* driving sugar metabolism [47,48] (**Fig 6D and 6E**). These results demonstrate robust associations between metabolic pathway activity, gene dependency, and drug response.

Lastly, we sought to integrate the results from Genetic PDEA and Pharmacological PDEA to identify metabolic pathway vulnerabilities that were consistent between gene dependency and drug response data (**Fig 6F and S8 Table**). By applying p-value and q-value filters to each analysis, we found three consistent vulnerabilities that were significant in both analyses. First, we found that when tyrosine metabolism is high, there is an increased vulnerability to inhibition or knockout of terpenoid backbone biosynthesis genes (**Fig 6G**). Interestingly, the drugs driving the enrichment (fluvastatin and pitavastatin) target the protein with the largest gene dependency within the pathway, HMGCR. We also found a common vulnerability between inactivation of the folate biosynthesis pathway and decreased aminoacyl tRNA biosynthesis (**Fig 6H**). Here, the top hits in Genetic PDEA and Pharmacological PDEA did not converge on a single protein product.

Nevertheless, these results indicate that inactivation of the folate biosynthesis pathway is more effective at slowing cancer cell growth when aminoacyl-tRNA biosynthesis pathway activity is low. Lastly, we found a strong association between inhibitors and gene knockouts of terpenoid backbone biosynthesis when pathway activity for biosynthesis of heparan sulfate is low (**Fig 6I**). Once again, the targets of the statins driving the Pharmacological PDEA enrichment did not align with the top gene dependencies (*DHDDS*, *HMGCS1*). Taken together, these results demonstrate common metabolic pathway vulnerabilities by integrating gene dependency, drug response, and gene expression data.

## Discussion

Traditionally, the analysis of gene essentiality in cancer cells has been limited to identification of individual genes required by cancer cells. Here, using metabolic pathways as an example, we have demonstrated an approach for identifying cancer cell dependencies at the level of pathways rather than individual genes. Illustrating the utility of our approach, we recapitulated known interactions between metabolic pathway activity, drug response, and gene dependency. These results build on a strong foundation of research identifying metabolic vulnerabilities in cancer cells [2,23]. Importantly, our results demonstrate that metabolic dependencies in cancer cells are highly context specific (**Fig 2**) and are impacted by the nutritional

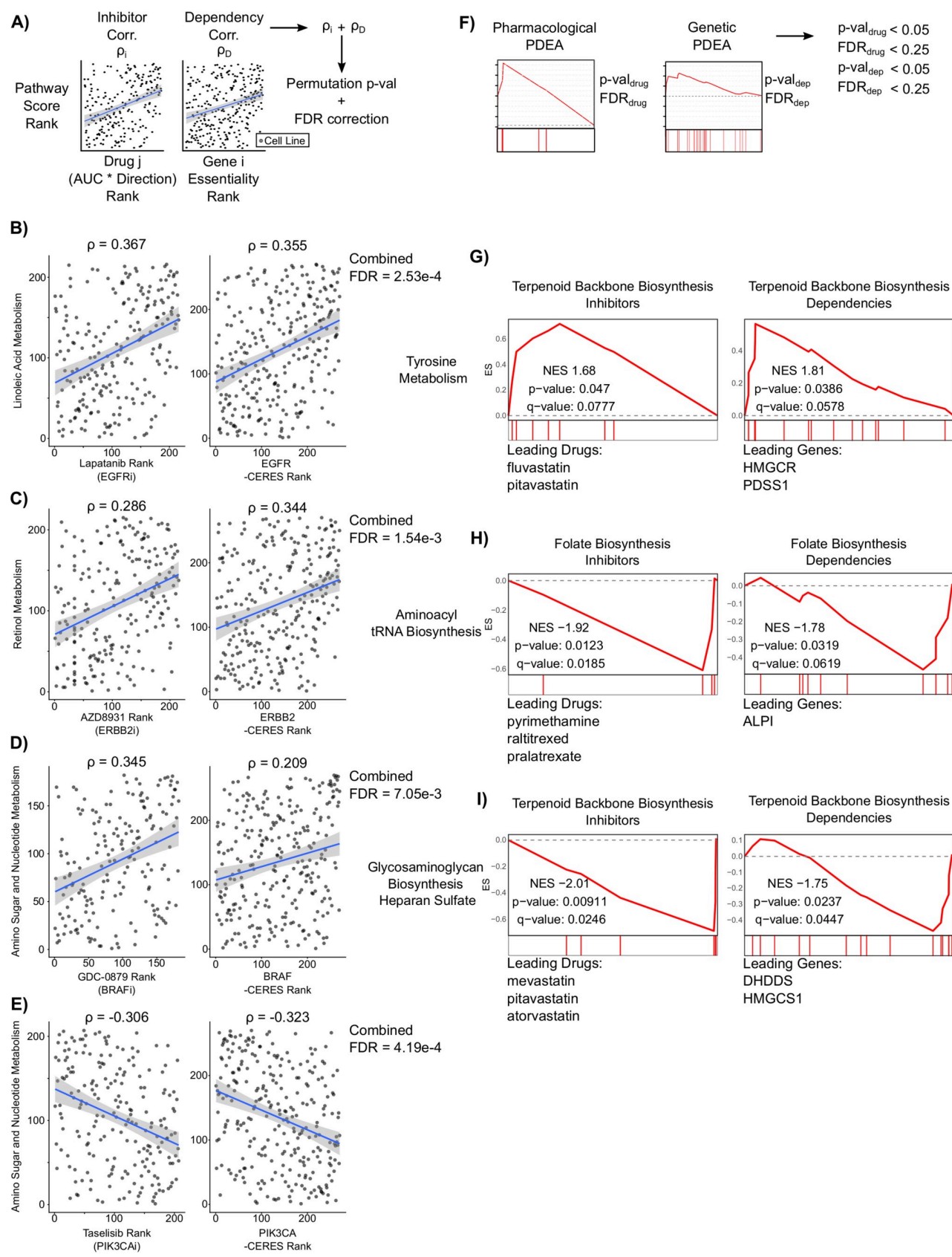

**Fig 6. Integration of pharmacological and genetic screens reveals consistent metabolic pathway vulnerabilities in adherent RPMI cell lines. A)** Schematic outlining approach to identify drug targets and genetic dependencies that are commonly increased or decreased with metabolic pathway activity. Significance was assessed by permutation testing combined with Benjamini-Hochberg FDR correction. 176 significant associations of 187,818 gene+drug:metabolic pathway combinations passed the FDR threshold of 0.01 (S7 and S8 Tables). **B-E)** Scatter plots of four drug response and CRISPR gene dependencies associated with metabolic pathway activity. The gene target of each drug is listed below the drug name. **F)** Schematic outlining a filtering approach used to identify common pathway-level vulnerabilities in Genetic PDEA and Pharmacologic PDEA. **G-I)** Mountain plots and leading edge drugs and genes from the three common pathway vulnerabilities are shown.

microenvironment (**Fig 3**). By investigating metabolic dependencies at the pathway level using genetic and pharmacological PDEA, we uncovered novel metabolic crosstalk, identified robust associations between drug response and metabolic pathway activity, and have discovered interactions between metabolic pathway activity and essentiality.

A recurring theme from our analyses is the importance of the pathways Folate Biosynthesis (hsa00790) and One-Carbon Pool by Folate (hsa00670). Folate metabolism supports two key metabolic phenotypes commonly found in cancer cells by producing one-carbon units for nucleotide synthesis and maintaining redox balance through production of NADPH [49]. In fact, two of the most widely used chemotherapeutics, methotrexate and 5-fluorouracil, target folate biosynthesis. In our analysis, we found that inhibitors of folate biosynthesis are highly effective when activity of other metabolic pathways is low (**Fig 5D**). This may be because the drugs classified as folate biosynthesis inhibitors are anti-metabolites that cannot be metabolized by enzymes like thymidylate synthase or dihydrofolate reductase. When adjacent metabolic pathway expression is low, compensatory mechanisms cannot offset decreased folate biosynthesis, causing a crisis in both nucleotide synthesis and redox homeostasis. Our results suggest that identifying biomarkers of activity for key pathways that control the sensitivity of antifolate pathways like the PPP (**Fig 5C**) would enable advances in patient selection for antifolate chemotherapy.

The approach outlined here also sets the stage for the use of metabolic pathways to guide patient selection to therapy. Patient selection for targeted therapeutics such as EGFR inhibitors is often based on mutations and copy number alterations (CNAs), but even for these targeted therapeutics there exists a need to identify additional features that inform patient selection. Here, we identified metabolic pathways that strongly correlate with both CRISPR knockout and pharmacological inhibition of the oncogenes EGFR, HER2, BRAF, and PIK3CA (**Fig 6**). This supports that metabolic pathways may be effective biomarkers even when mutations and CNAs in these oncogenes are not present (e.g., $\alpha$-linoleic acid metabolism for sensitivity to EGFR inhibitors). Additionally, within patient populations with mutations or CNAs in these oncogenes, metabolic pathway activity could serve as a biomarker that predicts response to therapy. Furthermore, our results support that the statins targeting HMG-CoA reductase (*HMGCR*) might be effective cancer therapeutics in tumors with high tyrosine metabolism and/or low heparan sulfate glycosaminoglycan biosynthesis. Taken together, our results demonstrate that there may be patient benefit in analyzing tumor metabolism to inform patient selection for targeted therapies.

Our analysis identified media composition as a major confounding factor when analyzing cancer cell metabolic pathway dependencies (**Fig 3**). This finding is consistent with recent studies and demonstrates the effect to which metabolism and metabolic vulnerabilities are shaped by the tumor microenvironment [23,50]. As such, our results highlight the importance of formulating cell culture mediums that better recapitulate the tumor microenvironment [51]. Furthermore, the data used in this study comes from adherent cell lines cultured on tissue culture plastic. This removes environmental stresses such as concentration gradients and physical stimuli that cells experience in real tumors. Recent efforts have demonstrated that

CRISPR-Cas9 screens can be performed in 3D organoids [50]. As this technology becomes more widely used, computational approaches such as ours can be applied to identify differences in metabolic pathway dependencies between 2D and 3D culture. Similarly, culture conditions that better reflect the physiological conditions of tumors will enhance the therapeutic relevance of our approach.

While our study identified robust associations between drug response and metabolic pathway expression, these analyses (**Figs 5 and 6**) rely on the annotated targets of these drugs. Off-target toxicity is a major concern when using small molecule inhibitors. In fact, some recent studies have found that off-target toxicity drives the anti-tumor effect of these compounds [25]. As such, we cannot exclude the possibility that off-target effects of these compounds could cause the associations identified here. Furthermore, some compounds in this study are quite promiscuous with multiple annotated protein targets. This promiscuity confounds the Pharmacologic PDEA analysis (**Fig 5**) since some drugs were mapped to multiple metabolic pathways. As such, the utility of the Pharmacologic PDEA approach lies in the aggregation of multiple drugs to arrive at a conclusion, rather than treating each individual drug as significant. In addition, by integrating the Pharmacologic PDEA and Genetic PDEA results (**Fig 6**), we strengthened our confidence in the association between metabolic pathway activity and pathway dependency.

Another potential weakness of our study is that we rely on the inference of metabolic pathway activity from gene expression data. Gene expression, however, does not always accurately reflect cellular metabolism. First, proteomic studies have shown that protein expression does not always correlate with gene expression [52]. Second, metabolic enzyme activity can be regulated by post-translational modifications [53,54]. By using gene expression data, we have not accounted for these factors, and as such our analysis may not reflect pathway activity at the metabolic flux level. We expect that expanding recent efforts to characterize metabolite abundance [33] and metabolite flux [55] in panels of cancer cell lines will improve our ability to identify metabolic pathway vulnerabilities by providing better measures of metabolic pathway activity.

Taken together, this study serves as a comprehensive characterization of the landscape of metabolic pathway vulnerabilities. Furthermore, our approach serves as a framework for integrating gene expression, gene dependency, and drug response data to uncover metabolic dependencies at the level of pathways rather than individual genes. We anticipate this approach could be extended to other biologically relevant pathways beyond metabolism. Furthermore, the utility of our approach will increase as CRISPR-Cas9 and pharmacologic screening expand to include more cancer cell lines, better measure of cellular metabolism, and physiologically relevant models like 3D organoids.

## Methods

### Data sources

Cancer cell line gene expression data was downloaded from the Cancer Cell Line Encyclopedia (CCLE) version 19Q4. Gene dependency data was downloaded from the Cancer Dependency Map (DepMap), Achilles gene effect version 19Q4 was used for the main figures in this study and the Sanger CRISPR (CERES) gene effect data [35] was used for the comparisons in S5 Fig. Drug response data was downloaded from the PRISM Repurposing database, version 19Q4 with secondary screen with dose response curve parameters was used. Metabolic pathway annotations were downloaded from the Kyoto Encyclopedia of Genes and Genomes (KEGG). We removed pathways that either 1) are not expressed in human metabolism or 2) contain less than five genes in the CCLE gene expression data set.

## Simulation studies

Gene expression data was simulated for 300 cell lines using a normal distribution for each cell line ($\mu = 0$, $\sigma = 0.5$). Then, a synthetic gene set of 25 genes was perturbed using a normal distribution gradient, where cell line 1 received a value randomly selected from a normal distribution with $\mu = -X$, $\sigma = 0.5$ and cell line 300 received a value randomly selected from a normal distribution of $\mu = +X$, $\sigma = 0.5$, with cell lines 2–299 receiving values randomly selected from normal distributions with sequentially increasing values $\mu$ from -X to X. Single-sample Gene Set Enrichment Analysis (ssGSEA) was calculated for the synthetic gene set for all 300 cell lines. Next, gene dependency data was simulated for the same 300 cell lines using the same normal distribution gradient method. For both gene expression and gene dependency data, values for X were varied from 0 to 0.5. Next, Spearman correlation coefficients between synthetic gene set activity (NES) and gene dependency were calculated for all 16,643 genes. Finally, Gene Set Enrichment Analysis was run to calculate the simulated Genetic Pathway Dependency Enrichment Analysis values. For the Pharmacological PDEA simulation study in S7 Fig, a similar approach was used with 200 cell lines and 1,390 drugs to simulate the data used in the Pharmacological PDEA study. For both simulation studies, the pipeline was run for 50 replicates.

## Calculation of metabolic pathway expression

1,019 cancer cell lines from the CCLE were separated by their culture type (adherent or suspension) and then culture medium (RPMI or DMEM), respectively. Cell lines with missing information for either culture type or medium were omitted. Gene expression values were unit normalized across all cell lines of the same culture and medium type (e.g. Adherent RPMI). Single-sample Gene Set Enrichment Analysis (ssGSEA) across all metabolic pathways in the KEGG database was run on the normalized gene expression values, giving normalized enrichment scores (NES) representing relative metabolic pathway activity for 69 metabolic pathways for each cell line. Normalized Weighted Average Expression (NWAS) was calculated as in [37]. Briefly, individual cell line gene expression values for each gene within a gene set are weighted by the corresponding number of pathways with which they overlap. Then, normalized weighted values are summed to achieve a pathway activity score.

## Genetic pathway dependency enrichment analysis

For each metabolic pathway, the NES was correlated with the -CERES score for all 16,643 genes. Due to the bimodal nature of NESs, Spearman correlations were used. The resulting correlation coefficients were ranked and GSEA querying KEGG metabolic pathways was run to calculate Genetic Pathway Dependency Enrichment Analysis (Genetic PDEA). Positive NES represent increased essentiality upon increased metabolic pathway activity, whereas negative NES represent increased essentiality upon decreased metabolic pathway activity.

## Drug response correlations

For each metabolic pathway, the NES was correlated with the -AUC (area under the curve) for 1,448 anti-cancer drugs in the PRISM repurposing database. Here, the AUC represents the dose dependent effect of a drug on cell growth, calculated by fitting a four-parameter logistic curve to viability values for each compound and cell line, with a lower AUC representing a stronger response to drug [21]. Drugs with less than 150 cell lines were removed, leaving 1,390 drugs. Spearman correlation p-values were calculated and a Benjamini-Hochberg false discovery rate correction was applied for each metabolic pathway.

### Pharmacological pathway dependency enrichment analysis

Drugs were mapped to their metabolic pathway using the annotated target(s) and genes from KEGG metabolic pathways. Since the PRISM database contains both activators and inhibitors, we annotated all activators by mechanism of action and multiplied their correlation coefficients by -1. Therefore, a pathway activator would be counted similarly to a pathway inhibitor. Pathways with 4 or more drugs were kept. Then, GSEA was run on the rank lists of 1,390 correlation coefficients.

### Integration of individual drug response and gene dependency

Drug-gene dependency pairs were mapped using the target annotations for each drug. Correlation coefficients for each drug and gene dependency were summed for each metabolic pathway, generating 187,818 drug+gene:pathway combinations. An empirical permutation test was run sampling 1,000 combinations of each drug+gene correlation coefficient. P-values were calculated by dividing the number of permutations that out-performed the real summed correlation coefficients by the number of same-signed permutations. P-values were then adjusted using a Benjamini-Hochberg correction.

### Integration of genetic PDEA and pharmacological PDEA

Results from Genetic PDEA and Pharmacological PDEA were filtered for same signed NES and p-values of less than 0.05 and FDR values of less than 0.25 (per the original GSEA algorithm). Three pathway-drug-gene dependencies were identified out of a possible 3,220 combinations.

### Supporting information

**S1 Fig. A normal distribution with mean of 0 and standard deviation of 0.5 reflects gene expression profiles.** Gene expression data was taken from the Cancer Cell Line Encyclopedia (CCLE) and scaled and centered within culture type (adherent or suspension) and culture medium (DMEM or RPMI). Four cell lines were chosen at random and their gene expression profiles (red) were compared to a normal distribution with a mean of 0 and a standard deviation of 0.5 (green).
(EPS)

**S2 Fig. Genetic PDEA in Adherent DMEM cell lines reveals context-specific pathway essentialities.** Metabolic pathway activity was inferred using single-sample GSEA (ssGSEA) for 153 adherent cell lines cultured in DMEM and correlated to gene dependency data from The Cancer Dependency Map (DepMap). Correlation coefficients were then ranked and GSEA was run querying the KEGG metabolic pathways (see Fig 1). **A)** Hierarchical clustering was performed on the Genetic PDEA normalized enrichment scores (NES). Results for pathways with FDR < 0.25 are plotted. Dots are colored according to their NES and sized according to the -$\log_{10}$ of the false discovery rate (FDR). Numerical values for each pathway can be found in S1 Table. **B)** Increased TCA cycle activity is associated with increased dependency on One-Carbon Pool by Folate metabolism.
(EPS)

**S3 Fig. Metabolic dependence on TCA cycle metabolism is context dependent.** In adherent RPMI cancer cells, when Glycolysis-Gluconeogenesis (hsa00010) pathway activity was low, the dependence on the TCA Cycle (hsa00020) was increased (top). When Pentose Phosphate Pathway activity (hsa00030) was high, the dependence on the TCA Cycle was increased (bottom).

The scatter plots of pathway activity NES and gene dependency (-CERES) for leading-edge genes *ACO2* and *SDHD* with Glycolysis-Gluconeogenesis and Pentose Phosphate Pathway, respectively, are shown.
(EPS)

**S4 Fig. Pathway activity does not correlate with pathway dependency.** Metabolic pathway expression was correlated with gene dependency data and GSEA was run on the resulting correlation coefficients (see Fig 1). Then, results were filtered for results that had the same pathway expression and pathway dependency tested (e.g. Glycolysis dependency GSEA was queried against all correlations with Glycolysis expression). **A)** The resulting normalized enrichment scores (NES) are presented as a density plot for both Adherent RPMI and Adherent DMEM analyses. The distribution of NES is centered around 0. **B)** The NES for each self-dependency is plotted for Adherent RPMI and Adherent DMEM. Values for each self-dependency can be found in S2 Table.
(EPS)

**S5 Fig. Comparison of Genetic PDEA for gene dependency data sets between the Broad and Sanger Institutes.** To examine the reproducibility of Genetic PDEA, we compared data from pan-cancer CRISPR-Cas9 gene dependency data set (Sanger Institute) through the Genetic PDEA pipeline. We combined statistical tests using the harmonic mean p-value (HMP). When applying an HMP threshold of 0.05 and a false discovery rate threshold of 0.05, we see 96% and 90% agreement between the Genetic PDEA results from the DepMap and the Broad Institute.
(EPS)

**S6 Fig. Normalized weighted average expression (NWAS) gives similar results to ssGSEA for Genetic PDEA.** To directly compare NWAS with ssGSEA, we re-ran our pipeline using NWAS to analyze dependency on the Pentose Phosphate Pathway. We chose the Pentose Phosphate Pathway for this comparison because several enzymes are shared between pathways (e.g., PRKL, PFKM, and PKFP are present in both Glycolysis-Gluconeogenesis and Pentose Phosphate Pathway gene sets). We found broad agreement between the metabolic pathway dependencies when using either NWAS or ssGSEA for both Adherent RPMI and Adherent DMEM cell lines (Spearman r of 0.606 and 0.691, respectively).
(EPS)

**S7 Fig. Pharmacological Pathway Dependency Enrichment Analysis Simulation Study. A)** Schematic representing the strategy used to integrate metabolic pathway activity with drug response screens. Like Genetic PDEA (Fig 1), the drug response (area-under-the-curve, AUC) for individual cancer cell lines was correlated to metabolic pathway activity as measured by ssGSEA. Drugs classified as activators (e.g., agonists) were multiplied by -1 for directional consistency. Cancer cell lines were separately processed by culture type and culture medium with a focus on adherent cell lines. All correlation p-values were FDR corrected using a Benjamini-Hochberg correction. Here, individual drug-pathway correlations are shown. Drugs were also mapped to metabolic pathways using their annotated gene targets and then Pharmacological PDEA was run on the resulting drug sets. Those results are presented in **Fig 5B and 5C) Simulated data (see Methods) was used to assess the sensitivity of the Pharmacological PDEA approach. Values added to the expression gradient resulted in slightly stronger correlation coefficients and Pharmacological PDEA results compared to values added to dependency gradient.
(EPS)

**S8 Fig. Pharmacological PDEA Reveals Metabolic Pathway Vulnerabilities in Adherent DMEM cell lines. A)** Pharmacological PDEA (see S4 Fig) was performed on 1,390 anti-cancer drugs from the PRISM database for 97 adherent cell lines grown in DMEM. Drugs were mapped to metabolic pathways by their annotated target(s). Hierarchical clustering was performed on NES and results with FDR < 0.25 are plotted. Dots are colored according to the NES and sized according to the $-\log_{10}$ FDR. **B)** Inhibitors of Folate Biosynthesis (hsa00790) are more effective when overall metabolic pathway expression is low in Adherent DMEM cell lines.
(EPS)

**S1 Table. Genetic Pathway Dependency Enrichment Analysis (Genetic PDEA) Results for Adherent DMEM and Adherent RPMI cell lines.**
(XLSX)

**S2 Table. Genetic Pathway Self-Dependency for Adherent RPMI and Adherent DMEM cell lines.**
(XLSX)

**S3 Table. Comparison of Genetic PDEA for gene dependency data sets between the Broad and Sanger Institutes.**
(XLSX)

**S4 Table. Individual Drug Response-Metabolic Pathway Expression correlations for Adherent RPMI and Adherent DMEM cell lines.**
(XLSX)

**S5 Table. Pharmacological Pathway Dependency Enrichment Analysis (Pharmacological PDEA) Results for Adherent DMEM and Adherent RPMI cell lines.**
(XLSX)

**S6 Table. Individual Gene Dependency and Drug Response—Metabolic Pathway Expression associations for Adherent RPMI cell lines.**
(XLSX)

**S7 Table. Individual Gene Dependency and Drug Response—Metabolic Pathway Expression associations for Adherent DMEM cell lines.**
(XLSX)

**S8 Table. Merged Genetic PDEA and Pharmacological PDEA results for Adherent RPMI and DMEM cell lines.**
(XLSX)

## Author Contributions

**Conceptualization:** James H. Joly.

**Formal analysis:** James H. Joly.

**Funding acquisition:** Nicholas A. Graham.

**Methodology:** James H. Joly, Brandon T. L. Chew.

**Software:** James H. Joly.

**Supervision:** Nicholas A. Graham.

**Visualization:** James H. Joly.

**Writing – original draft:** James H. Joly, Nicholas A. Graham.

**Writing – review & editing:** James H. Joly, Nicholas A. Graham.

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
