## [Decision Letter · Decision Letter 0]

26 Dec 2020

Dear Dr. Graham,

Thank you very much for submitting your manuscript "The landscape of metabolic pathway dependencies in cancer cell lines" for consideration at PLOS Computational Biology.

As with all papers reviewed by the journal, your manuscript was reviewed by members of the editorial board and by several independent reviewers. In light of the reviews (see below in this email), we would like to invite the resubmission of a significantly-revised version that takes into account the reviewers' comments.

As you think about whether and/or how to address the reviewer comments, we would advise you to consider the scope of this journal and its focus on computational biology in the context of some comments made by the reviewers that suggest you to perform experiments.  We think additional experiments may not be necessary.   

We cannot make any decision about publication until we have seen the revised manuscript and your response to the reviewers' comments. Your revised manuscript is also likely to be sent to reviewers for further evaluation.

Sincerely,

Jason W. Locasale, Ph.D.

Guest Editor

PLOS Computational Biology

Douglas Lauffenburger

Deputy Editor

PLOS Computational Biology

Reviewer's Responses to Questions

**Comments to the Authors:**

Reviewer #1: Manuscript title: The landscape of metabolic pathway dependencies in cancer cell lines

Summary:

The manuscript by Joly et al., describes an integration of genetic dependency data (DepMap CRISPR/CAS screens), gene expression data (RNAseq), and pharmacological drug sensitivity information (DepMap PRISM) on CCLE cell lines and provides a means for metabolic pathway-level interpretation. Their approach identifies novel and previously known links between metabolic pathway activity (expression levels) in a given cell line, and essentiality of genes in various metabolic pathways as well as sensitivity of cell lines to drugs targeting various metabolic pathways. Overall, they developed a useful analysis pipeline for interpretation of multi-level treatment data on cancer cell lines and provide valuable insights into future attempts for drug discovery, personalized medicine, and combination therapies. Overall, I think it is a solid work that fits well the scope and standards of the journal PLOS Computational Biology. I believe this manuscript is suitable for publication subject to the following revisions.

Major points:

1- Figure 3 legend is inconsistent with the text. Line 215 mistakenly states that folate concentration is 4 times higher in RPMI than DMEM, while the reverse is true and consistent with the main text (line 226). Also, line 216 in the legend mistakenly states dependency on Folate is higher in DMEM, while the reverse is true and consistent with the figure and main text.

2- Figure 3, right color bar, shows Folate twice, with different corresponding dependency values. Please correct.

3- Line 232: The authors attribute the difference seen between RPMI and DMEM in dependency to Oxidative phosphorylation to the fact that RPMI and DMEM contain 150 μM and 0 μM aspartate, respectively. They should add a plausible explanation as to why this difference is not also reflected in the dependency to Aspartate, Alanine, Glutamate metabolic pathway which also involves aspartate.

4- The authors need to add some discussion/clarification about the effects of media composition on dependencies. Many instances shown in Figure 3 are not consistent with the overall conclusions of this section (lines 234-236) claiming media composition affects pathway dependency. For instance, Glutathione, Histidine, and Phenylalanine are example of reagents with considerable differences in fold abundance between the two media conditions, but no apparent effect on relative dependencies. Overall, I think in order to be able to claim a general effect from media composition (or culture mode) on dependencies, a robust statistical method such as ANOVA should be used and appropriate p-values should be reported.

Minor points:

1- The authors used ssGSEA, which is a rank-based enrichment method, for inferring metabolic pathway activities based on expression levels of individual genes. The choice of this approach over alternative methods such as normalized weighted average expression should be justified either by direct comparison or at least by a logical explanation.

2- Throughout the manuscript, the authors only consider 69 of a total of 91 metabolic pathways defined by KEGG. The rationale behind this choice is not clear.

3- Line 153-154 is vague. Instead of using the word “association”, the authors should use “enrichment” (GSEA FDR<0.05).

4- Line 194: The authors raise an interesting question about whether there might be a link between a metabolic pathway’s essentiality and its own activity level. They report that of the 69 metabolic pathways queried, 36 had a negative NES and 33 had a positive NES. From this, they conclude that there is no general rule regarding pathway expression and essentiality. A question that bears asking is whether any meaningful differences could be detected between the pathways that fall into the above two group (corresponding to the 2 peaks shown in supp. Figure 3A). The authors should at least include the names of the pathways in each of the two groups in a supplementary table rather than only reporting the numbers 33 and 36 to facilitate further investigation beyond what is currently stated in the manuscript.

5- Line 244: Please describe how this AUC is measured and why IC-50 is not used instead.

6- Line 276: Typo: “activity can be associated with anti-cancer drug sensitivity”

7- I am curious to see if the method introduced in the present manuscript could be used to test the experimental findings by Gao et al. ( Dietary methionine influences therapy in mouse cancer models and alters human metabolism Nature volume 572, pages397–401(2019)) showing that methionine deprivation has a synergistic anti-cancer effect with 5-Fluorouracil therapy? I think this could potentially add an interesting validation to the present manuscript.

Reviewer #2: OVERVIEW:

In the manuscript, "The landscape of metabolic pathway dependencies in cancer cell lines", the authors describe a novel pipeline which attempts to draw a correlation between CRISPR-Cas9 loss-of-function essentiality screens, pharmacological, and gene expression data. Indeed, the authors were able to find correlations. The pipeline is poorly described and it is very difficult for the reader to find crucial details about the analysis. Also the need for the pipeline is not clearly delineated by the authors in the Introduction, discussion, methods or results. The authors have written a lengthy and redundant manuscript for example methods section of simulation studies is basically same as what they wrote in Results section. The authors apply their pipeline to sample data to show its sensitivity. However, no comparison between the sample data and real data is available. The pipeline described by the author is unclear to me; however, I feel like the analysis presented in the paper still needs more clarity and fleshing out. Further, the pipeline makes use of a lot of data only to recapitulate existing knowledge and very little new findings. The state of the manuscript still feels preliminary to me, in terms of both analysis and presentation, and leaves a lot to be desired. Keeping all this in mind, I recommend a major revision.

MAJOR CONCERNS:

1. The authors literally gave two paragraphs of background in the introduction before jumping to what they did in the paper. We often write Introductions in the paper to establish the relevance of the work and give the readers some context on where authors are coming from. Metabolic pathway dependencies in cancer are studied often and should have plenty of information for the authors to discuss. Please rewrite the entire section with some relevant examples as discussed in the results section. I would make some suggestions but I would review rather than write the paper for them.

2. The shape of the gene expression distributions of different cell lines may be significantly different. How similar are the gene expression distributions of simulated data from the real data (of the 16643 genes) for each of the 300 cell lines? Further, some of the cell lines are similar which the real data (both gene dependency and expression) will reflect. Do the authors introduce some "meaning" in their simulated data that their pipeline is supposed to catch? How do the authors ensure the pipeline will catch meaningful signals? Also, Could the authors provide a better way to understand 1B and 1C? What is the sensitivity of Genetic PDEA conveying? What does it mean? In line 118, what is a significant results here? The method can be sensitve, but can it capture accurate results? How do the authors know this? Perhaps some benchmarking and comparison of this simulated data would be useful for the reader to understand. The authors may have this in the supplementary information.

3. What does the metabolic pathway activity mean here? In Figure 1, what are the dimensions of sub-figure A1. The pipeline description is not clear.

4. If the authors are using DepMap dependency data which tells you the essentiality of the gene, why is there a need summarize the data at the level of pathway so early in the analysis. The authors could be loosing information by doing this. Can't we correlate the genes themselves and then finding the pathways with high positive or negative correlation coefficients.

5. Analysis in Fig 2, did that involve extraction before or after Genetic PDEA? Also how were genes belonging to multiple pathway treated?

6. At line 201, by context, do the authors mean environment (media)? Also, it should be obvious, no? that the pathway dependencies were influenced by media? A metabolic gene that is essential in one media doesn't need to be essential in another. However, it would be more curious if this were true for significant number of non-metabolic genes and pathways. Also, Could the authors show how much influence media has over the NES and correlation coefficients?

7. The way the whole analysis is done, the authors continue to integrate any new data they find. However, it would be nice if authors could validate some of their analysis. Alternatively, present new targets or novel unexplored pathway vulnerabilities, for e.g. using analysis in Fig 6. As of now, it seems like a method that is very expensive that needed transcriptomic data, gene essentiality data, and then pharmacological data, to produce known findings.

MINOR CONCERNS:

1. Line 101: Should that ssNES?

2. Line 133: "Then, a synthetic gene set of 25 genes was perturbed..." For better visualization of the perturbation, could authors also say how much does the value of each gene possibly change. Also what is X and its value?

3. Something doesn't see right with the sentence in line 159. Are authors trying to say "...dependecies are clustered together based on pathway activities more than activities are clustered based on dependencies..."? How? It is not clear from just looking at the figure 2A or Supp Fig 1?

Reviewer #3: The present study is intended to provide a comprehensive characterization of metabolic pathway vulnerabilities in cancer cell lines. The authors describe computational value scores of metabolic pathway activity and calculate associations between such scores and sensitivity to clinically approved drugs. They report that metabolic pathway dependencies are highly context-specific and that cancer cells are vulnerable to inhibition of one metabolic pathway only in conjunction with another specific metabolic pathway. As an example, they argue that their approach implemented for Pentose Phosphate Pathway may serve to identify which patients that respond to antifolate chemotherapies. Overall, while the study has potential value for future applications, the authors should provide at least some experimental evidence, for example test the effects of combinations predicted by the computational model. Such experiments should be done in various media and O2 concentrations to recapitulate tumor microenvironment heterogeneity. In absence of any experiment, it is difficult to judge the value of such computational predictions.

More specific concerns highlighted below:

1. Starting at line 264, the authors state that they “found a strong association between decreased Core Glycolysis (hsa_M00001) pathway activity and increased sensitivity to AZD8931, an inhibitor of EGFR and ERBB2. This should be a good test case in principle, but still unclear to me how it would be done exactly. For example treat a collection of cells with a combination of AZD8931 and inhibitors of various glycolytic enzymes? (how many and on which exact enzymes?). I presume more than one, otherwise it would undermine the foundation of the paper (the importance of considering the pathway rather than individual genes).

2. This reviewer remains unsure about the meaning (and usefulness) of metabolic pathway score/value as described. For example, in the case of cholesterol biosynthesis, I understand the essentiality score for HMGCR, or sensitivity scores for statins. When it comes to the entire pathway, my understanding is that the calculation is based on every component of the pathway. However, within any given pathway only a few steps are druggable and/or are rate limiting for the respective metabolic flux. To shut down an entire pathway one need only block a few select steps.

3. The reported correlations are based on datasets generated using CRISPR screens. The authors should consider recapitulating the analysis using the RNAi screen data (also available on DepMap). There are many cases of discrepancies between shRNA and CRISPR based screens (Avana, Demeter scores). In absence of any experimental approach, it would only be prudent to generate computational models using both types of data.

The figures depict CCLE collection as a “nondescript cloud” without further analyses based on lineages, key mutations, etc. Any correlations between the pathway score and individual scores for rate limiting enzymes?

**Have all data underlying the figures and results presented in the manuscript been provided?**

Reviewer #1: Yes

Reviewer #2: Yes

Reviewer #3: Yes

PLOS authors have the option to publish the peer review history of their article (what does this mean?). If published, this will include your full peer review and any attached files.

Reviewer #1: No

Reviewer #2: No

Reviewer #3: No
---

## [Decision Letter · Decision Letter 1]

6 Apr 2021

Dear Dr. Graham,

We are pleased to inform you that your manuscript 'The landscape of metabolic pathway dependencies in cancer cell lines' has been provisionally accepted for publication in PLOS Computational Biology.

Best regards,

Jason W. Locasale, Ph.D.

Guest Editor

PLOS Computational Biology

Douglas Lauffenburger

Deputy Editor

PLOS Computational Biology

Reviewer's Responses to Questions

**Comments to the Authors:**

Reviewer #1: I think the authors have adequately addressed my concerns and I have no further comments.

**Have the authors made all data and (if applicable) computational code underlying the findings in their manuscript fully available?**

Reviewer #1: Yes

PLOS authors have the option to publish the peer review history of their article (what does this mean?). If published, this will include your full peer review and any attached files.

Reviewer #1: **Yes: **Mahya Mehrmohamadi

---

## [Editor Report · Acceptance letter]

14 Apr 2021

PCOMPBIOL-D-20-01864R1 

The landscape of metabolic pathway dependencies in cancer cell lines

Dear Dr Graham,

I am pleased to inform you that your manuscript has been formally accepted for publication in PLOS Computational Biology. Your manuscript is now with our production department and you will be notified of the publication date in due course.

With kind regards,

Katalin Szabo
